# 3D AND-Type Stacked Array for Neuromorphic Systems

**DOI:** 10.3390/mi11090829

**Published:** 2020-08-31

**Authors:** Taejin Jang, Suhyeon Kim, Jeesoo Chang, Kyung Kyu Min, Sungmin Hwang, Kyungchul Park, Jong-Ho Lee, Byung-Gook Park

**Affiliations:** 1Department of Electrical and Computer Engineering, Seoul National University, Seoul 08826, Korea; star99@snu.ac.kr (T.J.); sh.kim@live.co.kr (S.K.); jess907@naver.com (J.C.); mkk2975@snu.ac.kr (K.K.M.); smh88@snu.ac.kr (S.H.); dremata104@gmail.com (K.P.); jhl@snu.ac.kr (J.-H.L.); 2Inter-university Semiconductor Research Center (ISRC), Seoul 08826, Korea

**Keywords:** neuromorphic, 3D AND-type synapse array, flash memory, Fowler–Nordheim tunneling

## Abstract

NOR/AND flash memory was studied in neuromorphic systems to perform vector-by-matrix multiplication (VMM) by summing the current. Because the size of NOR/AND cells exceeds those of other memristor synaptic devices, we proposed a 3D AND-type stacked array to reduce the cell size. Through a tilted implantation method, the conformal sources and drains of each cell could be formed, with confirmation by a technology computer aided design (TCAD) simulation. In addition, the cell-to-cell variation due to the etch slope could be eliminated by controlling the deposition thickness of the cells. The suggested array can be beneficial in simple program/inhibit schemes given its use of Fowler–Nordheim (FN) tunneling because the drain lines and source lines are parallel. Therefore, the conductance of each synaptic device can be updated at low power level.

## 1. Introduction

Recently, artificial neural networks (ANNs) have shown tremendous performance capabilities in a variety of areas, particularly in the areas of recognition, face detection and voice recognition. However, software-based ANNs, which use GPUs for vector-by-matrix multiplication (VMM), consume a large amount of power. On the other hand, neuromorphic systems based on spiking neural networks (SNNs) have received much attention owing to their low power consumption through event-driven operation. Among the elements of a neuromorphic system, synapses can easily implement VMM by simply summing the current. Various memristors such as phase change memory (PCM) and resistive memory (RRAM) have advantages in terms of scaling and given their simple structures, but there are certain limitations, such as reliability and sneak current issues [1,2,3,4,5,6,7,8,9,10].

Another synaptic device candidate is flash memory, which is quite mature and has the advantage of stable operation.in particular, NOR/AND-type flash memory has been studied in an effort to implement a synaptic array [11,12,13,14,15,16,17,18]. However, the cell size in this case exceeds that of memristors, and there are challenges when attempting to scale a large number of synapse arrays. For this reason, we proposed a 3D stackable AND-type synapse array architecture. In the proposed structure, the sources and drains (S/Ds) are formed by tilted implantation and conformal doping is possible if buffer oxide is deposited onto the top layer. Based on the above structure, program/inhibit pulses were applied to verify the successful control of the threshold voltage by means of a TCAD simulation.

## 2. Characteristics of the AND-type Synapse Array

In SNNs, upon the input of voltage, the VMM is realized by summing the current through each synaptic device. Therefore, the input/output lines of synaptic devices must be perpendicular, and such a structure is termed a NOR/AND-type array (Figure 1). The advantage of the NOR flash structure is that when the same input voltage is applied to the gate and drain during the inference process, it can be operated in an event-driven mode to minimize the leakage current. However, a hot carrier injection should be used to control the conductance levels of individual cells, and this can cause a great deal of damage to the silicon oxide, adversely affecting the reliability of the device. In addition, the program/erase cycle through a hot carrier injection has a disadvantage in that it consumes much more power compared to the FN tunneling method [19]. Alternatively, FN tunneling can be used with an asymmetric dual gate, but the process can be somewhat complicated [20,21].

Therefore, the device consumes less energy for weight updates in less time as well. Additionally, because the cell size decreases as cells are stacked, another advantage is realized when integrating a large number of synapses.

## 3. Results

### 3.1. Process Flow of the 3D AND-type Synapse Array

Figure 2 summarizes the process flow of the overall synapse array, simulated here using the Sentaurus process tool. First, SiO_2_ and in situ phosphorus-doped poly-Si are alternately deposited onto the stacked cells, and the active regions are patterned (Figure 2b). The basic structure is an AND-type array in which SLs and DLs are composed of doped poly-Si, with these two lines formed in parallel. Then, the doped poly-Si of the thin channel is removed through a chemical dry etching, after which poly-Si is deposited by chemical vapor deposition. This is done to minimize the leakage current by forming an intrinsic channel. Subsequently, dry etching is conducted to separate the cells of each layer, and a storage layer capable of storing the conductance of the synapse is deposited. Word lines are then formed in the vertical direction of each layer to transmit spiking signals to the devices on each layer. Double gates capable of controlling the conductance of cells and transferring the signals to post-neurons are formed at the side of each cell.

Generally, when implementing a stacked array, it is difficult to form S/Ds. Therefore, junction-less metal-oxide-semiconductor field-effect transistors (MOSFETs) are used for stacked arrays. However, the leakage current of a junction-less MOSFET is large such that current sum errors can arise at the SLs. Accordingly, we propose tilted implantation as a means by which to form S/Ds (Figure 2d). Tilted implantation has already been extensively studied in relation to the formation of S/D in the FinFETs [22,23,24,25,26,27,28,29,30,31,32,33]. However, several problems arose during the commercialization process related to the tilted angle of implantation. When the tilted angle is small, the differences in the dopant concentration at the top, center and bottom of the fin become large. Therefore, conformal doping is difficult, and the performance is therefore degraded. On the other hand, at greater tilted angles, reducing the gate pitch is limited due to the shadow effect. In the proposed structure, because the buffer oxide is located above the top cell, the vertical components of the implanted dopants are removed. Moreover, the horizontal dopants are uniformly injected into the cells of each layer, as shown in Figure 3.

Phosphorus at various tilted angles was simulated at a dose of 1 × 10^15^ cm^−2^ by a TCAD simulation (at an energy level of 20 keV). Figure 4 shows that the maximum dopant concentration is approximately 5 × 10^18^ cm^−3^ at 15° of tilt, which is a sufficient concentration for S/Ds. We analyzed the cell-to-cell variation in three layers at a tilt angle of 15° and confirmed that the dopant concentrations in the three layers did not differ significantly. Furthermore, the gate pitch can be reduced even further due to the low tilt angle.

Accordingly, it is possible to form a self-aligned S/D by tilted implantation, and rapid thermal annealing is then carried out at 950 °C for 10 s. after the metallization process, the 3D AND-type stacked array process is finally finished.

### 3.2. Cell-to-Cell Variation due to the Etch Slope

An etch slope of approximately 88° using an ICP poly-Si etcher with HBr gas can cause the cell-to-cell variation because it can affect the fin width and gate length. When the vertical cell pitch is 150 nm, the gate length and fin width increase by 5 nm as the cell goes down. Therefore, the on-current for the three layers was analyzed based on the upper cell, which had a gate length of 100-nm and a fin width of 100 nm, as shown in Figure 5a. Increasing the gate length and fin width has the effect of reducing the on-current, as shown in Figure 5b. In Figure 5c, we simulated the effects of both the gate length and the fin width on the current. As the gate length increases, the resistance of the channel also increases, which then reduces the current. Additionally, when the fin width increases, the spacing between the double gates increases, which reduces the gate controllability of the channel. Because both factors affect the current reduction, cell-to-cell variation occurs, with the current decreasing by approximately 4.7% each time the layer is lowered one-by-one.

There are two methods that can be used to resolve the issue of cell-to-cell variation caused by the etch slope as mentioned above. First, we can adjust the threshold voltage by applying more program pulses to the upper cells, as the current tends to decrease in the lower layer. However, in such a case, a crucial issue can arise because the threshold voltage of a large number of synapses must be set equally by applying a program pulse individually. The second method is to control the thickness of the lower layer to compensate for the low current. The thicker deposition of poly-Si in the lower cells is identical to increasing the effective width of the gate, implying that the current also increases. In addition, the cell-to-cell variation issue can be resolved simply, as the deposition thickness is easy to control.

### 3.3. Electrical Characteristics of Synapse Array

The proposed synapse array is an AND-type array structure; accordingly, individual conductance can be controlled through FN tunneling. The program and erase schemes are summarized in Table 1. When programming the selected cells, the FN tunneling condition is formed by applying 50 ns of a 13 V program pulse to the vertical word lines and 0 V to the SLs/DLs. Figure 6 shows the simulation results when applying ten program pulses to the selected cell, and it can be confirmed here that the current decreases as the threshold voltage shifts in the positive direction. For the unselected cells, the floating SLs/DLs inhibit FN tunneling by reducing the electric field at the tunneling oxide, which is a self-boost inhibit scheme (Figure 7) [15]. Because 0 V is applied to the word lines of the other inhibited cells, the threshold voltage of those cells does not change. When 100 inhibit pulses are applied to the unselected cells, the conductance does not change, and program disturb is well controlled. Using FN tunneling is highly cost-effective and convenient to control the conductance of numerous synapses, as the current is typically less than a few pA/µm.

Figure 8 shows a 4-bit operation of a synaptic device. Whenever a program pulse is applied to a target cell, the conductance gradually decreases, and it can be quantized to 16 levels. However, it is difficult to map the conductance linearly because with more program pulses applied to the cell, more charges fill the storage layer. In this case, the conductance can be more linearly quantized by means of incremental step pulse programming (ISPP) [34,35,36,37,38].

Next, we simulate the erase operation to increase the conductance of the synaptic device. After applying ten program pulses, a few erase pulses are applied to a programmed cell and the change of conductance is then analyzed. In Figure 9, −19 V is applied to the gate and 0 V is applied to the S/D for 1 ms as an erase pulse. During the erase operation, trapped electrons are removed and holes are injected into the storage layer, resulting in an increase in the positive net charges. As the erase pulses are applied, the number of trapped electrons decreases, which degrades the erase efficiency. Through these program and erase operations, the conductance values of individual cells can be mapped through FN tunneling.

Read disturb as well as program disturb can change the conductance of a synaptic device, reducing its accuracy. When implementing a synapse array with a NAND-type array, a pass voltage must be applied to de-selected cells of the same string during the inference operation, causing a read disturb (Figure 10a) [39,40,41,42,43,44,45,46,47,48,49]. However, in the proposed structure, there is little risk of a read disturb because there is no need to apply pass voltage to the word lines of de-selected cells (Figure 10b). Moreover, the read voltage is very low at 1 V, and the conductance does not change even if read pulses at a rate of more than 10^5^ times are applied. This makes the device more resistant to read disturb than a NAND-type array, and it has retention and endurance advantages compared to a NOR-type array, which uses hot carrier injection [50,51,52,53].

The retention characteristics are analyzed at room temperature after applying ten program pulses with different program voltages. When 13 V is applied to the selected cell, the electric field caused by trapped charges is too small to degrade the retention. The conductance changes by approximately 2.1% over ten years, but if the program voltage increases, the retention characteristics worsen, as shown in Figure 11. Therefore, we selected 13 V for the program voltage and thus obtain better retention characteristics.

## 4. Conclusions

We proposed a 3D AND-type stacked array to implement a synapse array. Tilted implantation can be used to form conformal S/Ds of individual cells by depositing a buffer oxide onto the top of the cell. Additionally, the cell-to-cell variation caused by the etch slope can be reduced by thickly depositing the cells in the lower layers. It was verified through a simulation that the conductance levels of individual cells could be adjusted gradually at a low power using FN tunneling. In addition, the 10-year retention characteristic can be guaranteed through a low program voltage.

## Figures and Tables

**Figure 1 micromachines-11-00829-f001:**
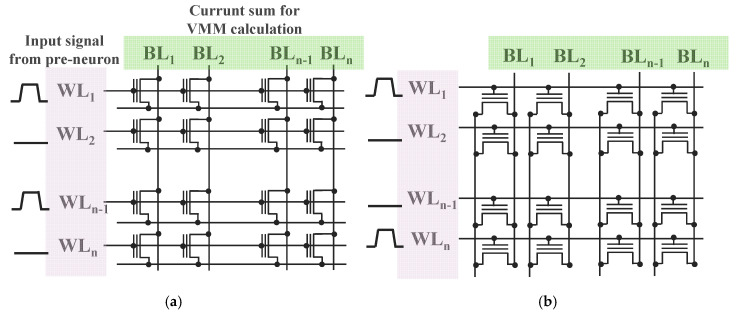
(**a**) NOR-type flash array structure. Because the directions of word lines and drain lines are identical, event-driven operation is simple if applying the same input pulses to those lines. (**b**) 3D AND-type stackable array with parallel source lines (SLs) and drain lines (DLs). The advantage of this structure is that the program and erase scheme is simple because the conductance can be adjusted through FN tunneling.

**Figure 2 micromachines-11-00829-f002:**
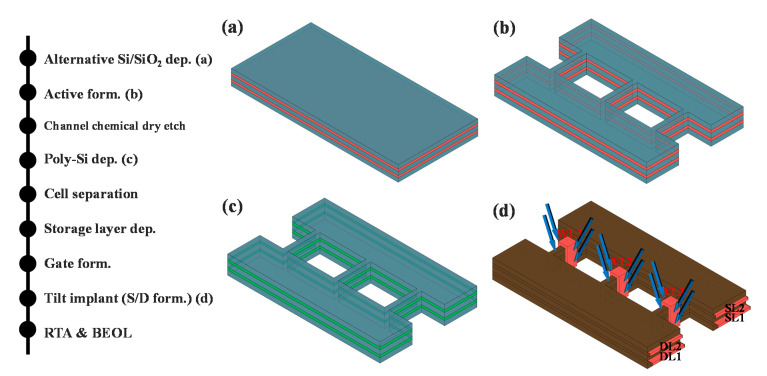
Process flow of the 3D AND-type synapse array (**a**) n+-doped poly-Si and SiO_2_ are alternatively deposited; (**b**) Active regions are formed by dry etching. The SLs/DLs are positioned in parallel; (**c**) intrinsic poly-Si is deposited to make channels. These channels can reduce the leakage current caused by the drain bias; (**d**) Self-aligned sources and drains are formed by means of tilted ion implantation. Phosphorus at a dose of 1 × 10^15^ cm^−2^ is implanted at slight tilt angle.

**Figure 3 micromachines-11-00829-f003:**
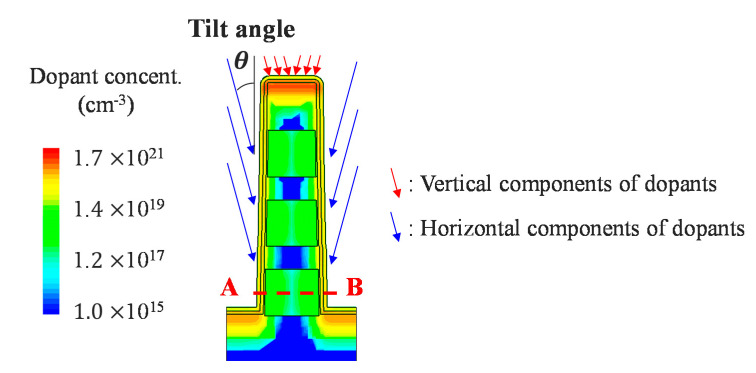
Dopant concentration after tilted implantation with a tilt angle of 15°. The vertical components of the dopants are implanted on the buffer oxide and are unable to affect the concentration of the cells. Because horizontal components are injected into each source and drain, this leads to similar doping concentrations among the cells.

**Figure 4 micromachines-11-00829-f004:**
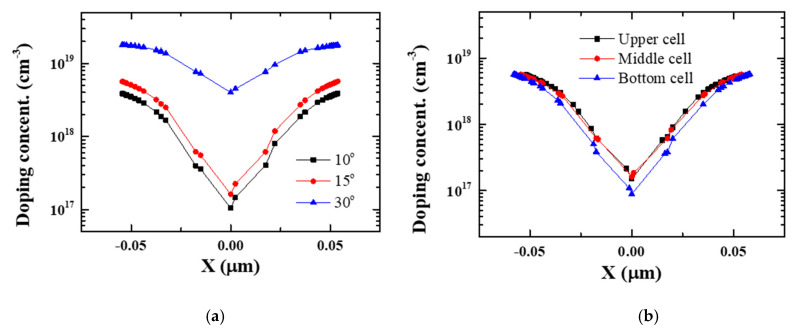
Doping concentration according to cut line shown in Figure 3: (**a**) Implantation with θ = 10°~15° is appropriate for the formation of SLs/DLs because the shadow effect is suppressed and a sufficient amount of dopant is implanted to SLs/DLs; (**b**) There are scant differences in the cell-to-cell dopant concentrations of the sources and drains.

**Figure 5 micromachines-11-00829-f005:**
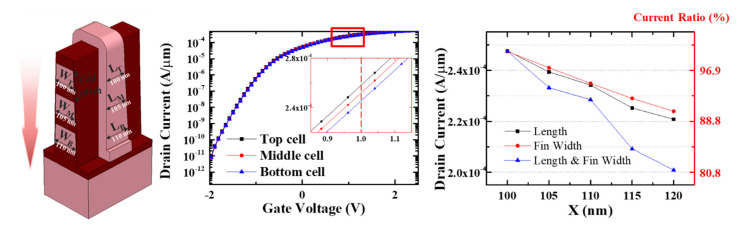
(**a**) Fin width and gate length of the lower cells increase due to the poly-Si etch slope; (**b**) The current of the lower cells tend to decrease when 1 V is applied to the gate; (**c**) The longer gate length increases the channel resistance and the wider fin width degrades the gate controllability. If the gate length and fin width become 120 nm, the current decreases by approximately 18.83% compared to when these values are both 100 nm.

**Figure 6 micromachines-11-00829-f006:**
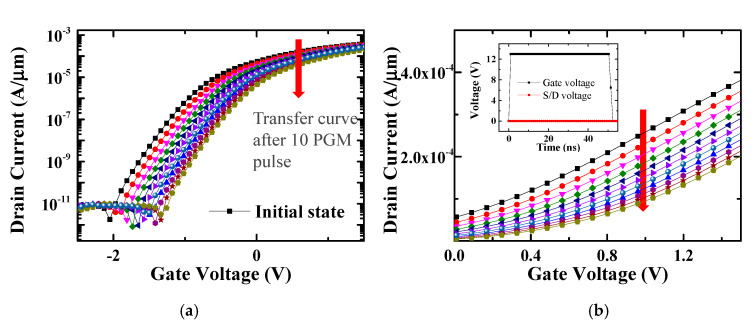
Transfer curves after applying several program/erase pulses: (**a**) The threshold voltages of selected cells increase as electrons become trapped in the trapping layer; (**b**) The transfer curves on a linear scale with different numbers of program pulses are shown.

**Figure 7 micromachines-11-00829-f007:**
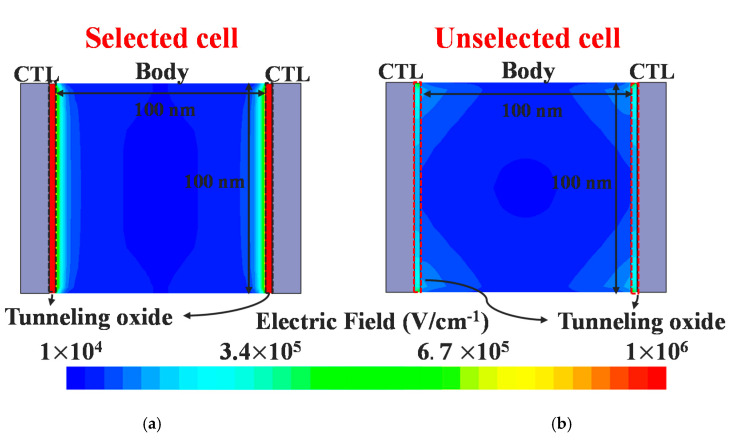
Contour plot of an electric field with the voltage condition shown in Table 1: (**a**) For the selected cells, the electrons are trapped in the storage layer by FN tunneling due to the high electric field of 1 × 10^6^ V/cm^−2^; (**b**) When the SLs/DLs are floating, the electric field of the tunneling oxide is too small to lead to FN tunneling owing to the self-boosted channel.

**Figure 8 micromachines-11-00829-f008:**
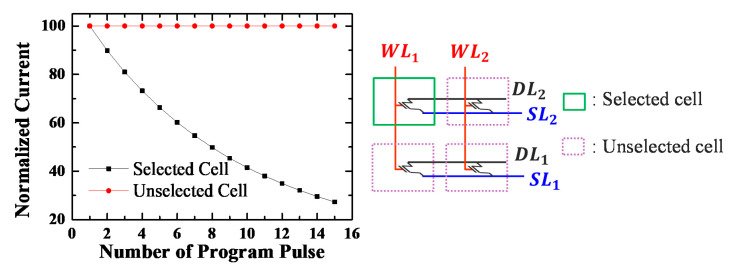
The normalized current of a selected cell decreases when applying 15 program pulses. On the other hand, the current levels of unselected cells in these three cases do not change. Each synapse can store 4-bit conductance.

**Figure 9 micromachines-11-00829-f009:**
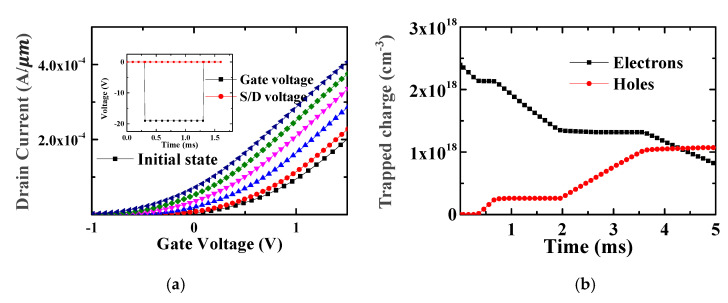
(**a**) When applying five erase pulses, the trans-characteristic curves are analyzed; (**b**) During the erase operation, positive net charges increase.

**Figure 10 micromachines-11-00829-f010:**
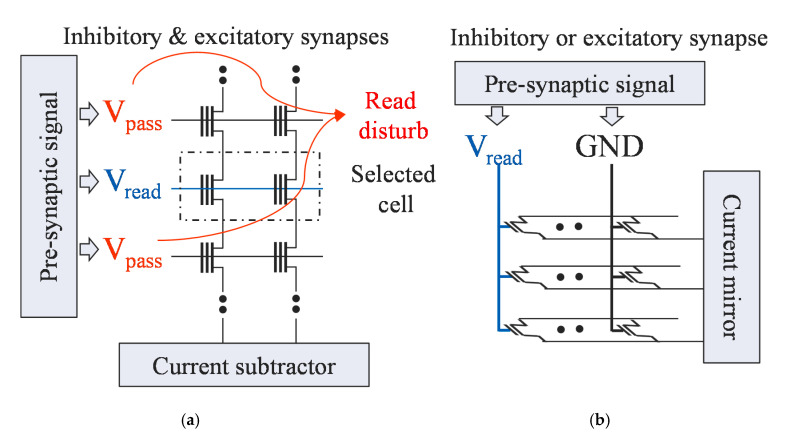
(**a**) Schematic of neural networks using NAND flash memory [40]; (**b**) Schematic of the proposed structure: pass voltage is not used on the de-selected cells and read disturb error can be suppressed.

**Figure 11 micromachines-11-00829-f011:**
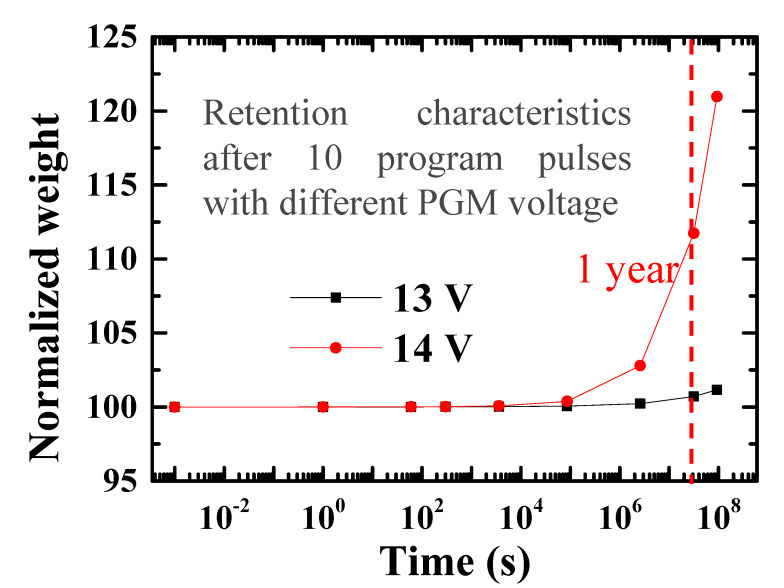
Retention characteristics according to the program voltages. After one year, the conductance programmed at 14 V changes by approximately 9.7%, whereas the change of the conductance programmed at 13 V is within 1%.

**Table 1 micromachines-11-00829-t001:** Program/erase scheme for changing conductance of the synaptic device.

	Program	Erase
Selected WL	13 V	−19 V
Unselected WL	0 V	0 V
Selected SL/DL	0 V	0 V
Unselected SL/DL	Floating or *V*_pgm_/2	Floating or *V*_er_/2

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
