# Peer review of "3D AND-Type Stacked Array for Neuromorphic Systems"

_micromachines, 2020, doi:10.3390/mi11090829_

Round 1

Reviewer 1 Report

The paper presents a 3D stacked AND type Flash array for VMM computation. Fabrication and device properties are studied using TCAD simulations. I-V curves are simulated for the bitcells.

Q1. Can you capture the effects of program and read disturbs in your simulations? What are the parasitics considered in the modeling of Fig 8 simulations?

Q2. Fig 8 shows only incremental decrements in weights. Can you should the other way around - increase in weights? 

Q3. Can you predict from your simulations - number of bits that can be stored, retention of multi-level states? 

There are several grammatical and compositional errors such as:

26: particularly in recognition (?), face detection and voice recognition. 

28: spiking neural networks (SNNs) has been got attention because of

Fig 6. tunneling oxide is too small to occur FNT

Fig 7: cells become increase

130: the threshold voltage of those cells does not be changed

Author Response

Response to Reviewer 1 Comments

Point 1: Can you capture the effects of program and read disturbs in your simulations? What are the parasitics considered in the modeling of Fig 8 simulations?

Response 1: Thank you for your comments. I simulated program disturb at Fig. 8. We simulated that the threshold voltage does not be changed for inhibited cells in three cases. And add the sentence at 136 for controlling the program disturb. As there is no need to turn on other cells to read the selected cells like a normal NAND flash, the effect of read disturb does not be great either. Figure 8 shows the device simulation for controlling the weight. Therefore, it cannot reflect the parasitic capacitance of the circuits. It is necessary to think about this at the system level, and it is planned to be done as a future work.

Point 2: Fig 8 shows only incremental decrements in weights. Can you should the other way around - increase in weights? 

Response 2: I add the erase operation and erase pulse scheme at figure 7. The conductance change is plotted when erase pulses of – 19 V are applied for 1 ms. Therefore, we show both of the increase/decrease in weights for transfer learning.

Point 3: Can you predict from your simulations - number of bits that can be stored, retention of multi-level states? 

Response 3: I simulated more simulation for 4 bit operations. Therefore, we apply 15 program pulses and that is shown in Fig. 8. If ISPP scheme is used for multi-bit operation, the weight can be quantized more gradually. Retention characteristics are added in Fig. 9. Because program voltage is small and pulse duration is short, the retention time is enough if using the 13 V of PGM voltage. (line at 143~146 and figure 9.)

Point 4: There are several grammatical and compositional errors such as: 

26: particularly in recognition (?), face detection and voice recognition. 

28: spiking neural networks (SNNs) has been got attention because of

Fig 6. tunneling oxide is too small to occur FNT

Fig 7: cells become increase

130: the threshold voltage of those cells does not be changed

Response 4: The journal was completely revised by requesting professional English correction. For example,

26 : particularly in the areas of recognition, face detection and voice recognition.

28 : spiking neural networks (SNNs) have received much attention owing to their low power consumption

Fig. 6. : the electric field of the tunneling oxide is too small to lead to FN tunneling owing to the self-boosted channel

Fig. 7. : The threshold voltages of selected cells increase as electrons are trapped in the trapping layer

130 : the threshold voltage of those cells does not change

Reviewer 2 Report

The paper is well written but I encourage a review by an english mother tongue. Unfortunately, the paper is just based on simulations. Experimental results would make the paper much more interesting. When dealing with memory arrays, integration results could be very different from simulation results. 

Author Response

Thank you for reviewing my paper.

The journal was completely revised after foreign language correction.

As you pointed out, we plan to expand the simulation to the array level and proceed with the actual process. Once again, thank you for your detailed review.
